# Peripheral Artery Disease Diagnosed by Pulse Palpation as a Predictor of Coronary Artery Disease in Patients with Chronic Kidney Disease

**DOI:** 10.3390/jcm12185882

**Published:** 2023-09-10

**Authors:** Daniel B. C. Dos Santos, Luis Henrique W. Gowdak, Elias David-Neto, Felizardo A. Nataniel, José J. G. De Lima, Luiz A. Bortolotto

**Affiliations:** 1Heart Institute (InCor), Hospital das Clínicas, University of São Paulo Medical School, São Paulo 05403-000, Brazil; luis.gowdak@gmail.com (L.H.W.G.); felizardonataniel@gmail.com (F.A.N.); jose.lima@incor.usp.br (J.J.G.D.L.); hipbortolotto@gmail.com (L.A.B.); 2Hospital das Clínicas, University of São Paulo Medical School, São Paulo 05508-090, Brazil; elias@cntt.com.br

**Keywords:** chronic kidney disease, peripheral artery disease, coronary artery disease, pulse palpation, doppler ultra sound

## Abstract

There is a need of simple, inexpensive, and reliable noninvasive testing to predict coronary artery disease (CAD) in patients with chronic kidney disease (CKD), where the prevalence of cardiovascular (CV) events and death is elevated. We analyzed the association between peripheral artery disease (PAD) and CAD in 201 patients with stage 5 CKD on dialysis using a prospective observational cohort. Diagnosis of PAD by both palpation and USD were significantly correlated. In patients with PAD diagnosed by palpation, CAD was observed in 80%, while in those diagnosed by USD, CAD was present in 79.1%. The absence of a pulse by palpation predicted CAD with a sensitivity of 55% and a specificity of 76%; USD showed a sensitivity of 62% and specificity of 60% to predict CAD. The risk of combined serious CV events and death was significantly higher in subjects with PAD diagnosed by palpation, but not by USD. PAD assessed by palpation also correlated with the occurrence of multivessel CAD and with the probability of coronary intervention. Both methods are moderately useful for predicting CAD, but PAD diagnosis by palpation was a better predictor of combined CV events and death and was also associated with CAD severity and likelihood of intervention.

## 1. Introduction

Peripheral artery disease (PAD) is a manifestation of systemic atherosclerosis often observed in connection with chronic kidney disease (CKD) [1]. PAD in the general and dialysis populations is a predictor of major cardiovascular events, especially those associated with coronary artery disease (CAD) [2,3]. CAD is highly prevalent and one of the main causes of morbidity and mortality in patients with CKD [2,4]. The exact proportion of dialysis patients that have both CAD and PAD is not well known. Also, there are few data on the influence of PAD on the prognosis of patients on dialysis with and without CAD.

The Issue of “to screen or not screen” CAD in an asymptomatic kidney transplant candidate is still controversial due to the lack of trials comparing conservative vs. invasive CAD management in patients with advanced CKD [5]. Recently, the ISCHEMIA-CKD trial showed that a routine invasive strategy was not associated with a reduction in cardiovascular events in 777 patients with CKD, the median age of 63 years old, and at least moderate ischemia [6]. In a study conducted by our group in patients older than 65 on hemodialysis, revascularization procedures did not influence the incidence of adverse cardiovascular events [7]. Thus, age might limit the benefit of revascularization procedures in older patients with stage 5 CKD because of their short life expectancy. In other work on 535 CKD patients on dialysis, we investigated whether pretransplant identification of CAD is helpful for defining prognosis and whether preemptive coronary intervention reduces the incidence of cardiovascular events and death after engraftment [8]. We found that although coronary assessment identified patients at increased risk of posttransplant coronary events, it did not differentiate between the risk of death in patients with and without significant CAD. Survival was similar in patients undergoing either medical or interventional treatment for CAD.

Four recent works discussed the usefulness of non-invasive stress studies for the assessment of patients being considered for renal transplantation [9,10,11,12]. All concluded that those testings had limited utility in that regard. It was also speculated that screening for CAD may contribute to delays in listing patients for transplant [9]. On the other hand, we observed that clinical stratification helps to identify patients who may benefit from non-invasive coronary evaluation [13]. Risk stratification based on myocardial stress testing was useful only in patients with an intermediate risk for events, but failed in patients at low and high risk. Therefore, there is a need for inexpensive, simple, and reliable methods for risk stratification that may reduce the number of unnecessary testings currently in use.

Given the prevalence and importance of PAD, it should be routinely investigated in patients on hemodialysis. There are several methods used for diagnosing PAD. The “gold standard” is angiography, which is indicated for those patients in whom a non-invasive test suggests significant vascular obstruction or when there is clear clinical evidence of severe disease [14,15]. Among the non-invasive methods, arterial Doppler ultrasound (USD) stands out as the most used due to its relatively low cost and availability in most services [16].

At our institution, we initiated a systematic prospective cardiovascular evaluation, following a pre-specified protocol, of patients on the kidney transplant waiting list. The objective was to define the best approach for the diagnosis and treatment of cardiovascular diseases (the KiHeart cohort), with an emphasis on CAD. Although the use of this approach has resulted in a significant decrease in the incidence of cardiovascular complications over the years, the cost and time taken to complete the assessment of patients at higher risk are considerable, as it requires the use of non-invasive tests for myocardial ischemia for most patients and coronary angiography in selected cases. Based on these findings, we have sought a methodology based on history and physical examination that is sufficiently accurate to allow the indication of coronary angiography without resorting to other non-invasive tests.

The purpose of this investigation was to determine the association between PAD, as assessed by noninvasive testing, and CAD in patients treated by hemodialysis while on the waiting list for renal transplantation and to assert the influence of that association on the prognosis and clinical management.

## 2. Materials and Methods

### 2.1. Patient Selection

The study was approved by the institutional ethics committee and conducted according to the Declaration of Helsinki. This was a single-center observational study on data collected prospectively from patients with CKD, stage 5, undergoing hemodialysis. Patients were being considered to receive their first kidney graft at the Renal Transplant Unit, Division of Urology, University of São Paulo Medical School (São Paulo, Brazil), and they were referred to the Heart Institute (InCor) (São Paulo, Brazil) for cardiovascular assessment before being formally included on the waiting list. Between January 2015 and March 2021, 246 patients were evaluated. Individuals who underwent coronary intervention before being admitted to the cohort and who had incomplete medical records were excluded, leaving 201 subjects to be finally selected for the study.

Patients were treated by maintenance hemodialysis performed in 4 h sessions, 3 times/week, with a bicarbonate bath and were maintained on statins, aspirin, ACE inhibitors (Angiotensin Converting Enzyme Inhibitors), and beta-blockers, regardless of symptoms or evaluation results, according to current guidelines for secondary prevention of cardiovascular events [17].

Patients were followed up prospectively for a period of 6 years and were evaluated at the beginning, after 12 months, and annually up to the conclusion of the study. Censored events were verified during clinical visits, by telephone, or by e-mail.

The primary endpoints were the composite incidence of major cardiovascular events (myocardial infarction, unstable angina, sudden death, stroke, peripheral vascular event, or new-onset heart failure), coronary events (unstable angina, myocardial infarction, and sudden death), and death from any cause.

### 2.2. Study Protocol

A prespecified comprehensive cardiovascular investigation was performed, as has been reported [16]. Patients underwent a 12-lead resting EKG and a transthoracic echocardiogram as part of their evaluation. Noninvasive testing for CAD with dipyridamole/adenosine myocardial stress testing by SPECT with Tc-99m Sestamibi was performed in selected cases (*n* = 186). Patients with noninvasive testing suggestive of CAD, diabetes, or clinical evidence of cardiovascular disease (angina, previous myocardial infarction or stroke, left ventricular dysfunction, or extracardiac atherosclerosis) were eligible for coronary angiography (*n* = 157). Significant CAD was arbitrarily defined as luminal stenosis ≥70% in one or more epicardial arteries or a 50% narrowing of a main left coronary artery by visual estimation from two independent experts. PAD was defined as either absence of a pulse in at least one of the main distal arteries of the lower limb (anterior and posterior tibial arteries) according to the same investigator or at least a 50% narrowing on the same vessels, as assessed by arterial Doppler ultrasound performed by an examinator blinded on the clinical vascular condition of the patient using the standard methodology routinely applied in our institution [16].

### 2.3. Statistical Analysis

The analysis was performed with a statistical program (R Core Team, version number: 2021, Vienna, Austria). Results were presented as means ± SD or percentages. Differences between groups were assessed with Fisher’s exact test (for categorical data), or the two-tailed Student’s *t*-test (for continuous data), or the Mann–Whitney test for independent samples, when appropriate. For the palpation method and USD of the lower limbs, performance measures with 95% confidence intervals were considered. To evaluate the performance of both methods, a value above 70% was considered. The correlation of variables was performed using Pearson’s or Spearman’s test accordingly.

Logistic regression analysis was used to investigate the association of PAD/CAD and outcomes. To determine independent predictors of complications, the univariate binary logistic regression model was used, considering the clinical variables that could be associated with such outcomes. The multiple binary logistic regression model was applied using the ENTER method, including in the regression model the variables of clinical interest, and with a *p*–value < 0.05 resulting from the univariate model, respecting the minimum limit of 10 events per variable included in the model. The factors most significantly associated with the occurrence of complications were presented as the Adjusted Odds Ratio (ORa) accompanied by a 95% confidence interval (CI).

The analysis of coronary event-free survival and death from any cause was performed using the Kaplan–Meier method, and the curves were compared using log-rank. The Cox model was used to assess variables independently associated with outcomes. Statistical significance was set at *p* < 0.05, and all *p* values are bilateral.

## 3. Results

The study population included 201 participants, 58.7% men and 74.6% white. The mean age was 55.2 (±11.8) years. Patients were on hemodialysis for 40.2 ± 45.6 months (median 13 months). Most individuals had at least one classic cardiovascular risk factor, including previous/current smoking (52.2%), dyslipidemia (32.3%), diabetes (56.7%), hypertension (57.7%), and overweight (mean body mass index: 27.1 ± 5.3 kg/m^2^). Other cardiovascular diseases were observed, such as previous stroke (9.5%), myocardial infarction (14.4%), and heart failure (10.9%). In the totality of the patients, PAD by absent palpation was associated with diabetes and hemodialysis time. Age, male sex, diabetes, and HDL values showed significant correlations (*p* < 0.05) with PAD diagnosed by USD (Table 1).

PAD was observed in 31% of patients using palpation of the pulses and 43% using the Doppler method. The diagnoses of PAD by palpation and by ultrasound were significantly correlated (*p* < 0.0001, OR = 27.37 95%CI 12.00–71.52). The proportion of individuals with absent pulses among those with obstruction ≥ 50% on Doppler was 64%. The specificity, sensitivity, positive predictive value, and negative predictive value of the palpation method for PAD, using USD Doppler as a reference, were the following: 94%, 64%, 89%, and 78%, respectively.

The prevalence of angiographically proven significant CAD in 157 patients undergoing the test was 68.1%. The proportion of individuals with an absent pulse who had CAD was 80% (OR = 2.58 95% CI 1.24–5.65, *p* = 0.014). CAD was observed in 79.1% of patients with PAD through the Doppler method (OR = 2.89 95% CI 1.45–5.93, *p* = 0.003). Regarding the specificity, sensitivity, positive predictive value, and negative predictive values of the palpation method and Doppler in the prediction of CAD, both methods showed low accuracy, although both had positive predictive values greater than 70% (Table 2). The results of myocardial scintigraphy, a screening method for CAD used in our service to indicate invasive coronary studies, did not correlate with the presence of CAD assessed by coronary angiography (OR = 1.71 (95% CI 0.87–3.45), *p* = 0.212). The specificity, sensitivity, positive predictive value, and negative predictive value for myocardial scintigraphy in the detection of CAD also showed low accuracy (60%, 53%, 73%, 38%, respectively).

Figure 1A,B show the Kaplan–Meier plots for the risk of combined cardiovascular events in patients with and without PAD by palpation and Doppler. The cumulative risk of cardiovascular events was significantly higher in patients with PAD assessed by the palpation method (*p* = 0.024), but was similar in patients with PAD defined by Doppler. The results suggest that detection of PAD by the absence of pulses is more efficient in detecting patients at higher risk of events. Figure 1C,D show the Kaplan–Meier plots for the risk of coronary events according to the palpation method and by Doppler. The results of the two methods did not correlate with the risk of coronary events. Figure 1E,F show the Kaplan–Meier plots for PAD-related all-cause mortality using the two diagnostic methods. The risk of death from any cause was significantly higher in individuals with PAD diagnosed by palpation (*p* < 0.001).

The impact of PAD on the prognosis, adjusted for other risk factors, is represented in Table 3 and Table 4. PAD, diagnosed by palpation, was a predictor of combined cardiovascular events and death from any cause after adjustment for other risk factors. The same, however, did not occur when the diagnosis was based on Doppler.

There was a significant association between PAD diagnosed by palpation and the occurrence of multivessel coronary disease, defined by stenosis in at least two epicardial arteries (*p* = 0.013) and with the probability of an indication for coronary intervention (*p* = 0.019). These associations were not verified when PAD was diagnosed by Doppler (*p*= 0.092 and 0.060, respectively).

## 4. Discussion

This study evaluated the association between PAD and CAD in patients on hemodialysis and the possible implications of this finding for the prognosis and management of patients with stage 5 CKD. The investigation exposes the complexity of screening for CAD in patients with chronic kidney disease with multiple risk factors, in particular, those undergoing dialysis treatment. Although coronary atherosclerosis is prevalent in this population, its symptoms are absent or atypical, which requires the indication of screening tests in most cases. As demonstrated in this study, myocardial scintigraphy showed low accuracy in the diagnosis of CAD for this population. Therefore, it is important to find diagnostic alternatives with adequate degrees of precision and that reduce the time, risk, and cost of the investigation [14,18].

We propose an alternative method for the prediction of CAD and the indication of coronary angiography: the assessment of PAD by the palpation method (absence of peripheral pulses). Two reasons justify our choice, the first because its presence correlates with CAD in the general population, and the second because it is possible to diagnose it by a simple palpation of the peripheral pulses, a maneuver that is part of the standard physical examination of every patient suspected of having a cardiovascular disease. We considered Doppler with stenosis ≥ 50% as the gold standard, as we were interested in verifying whether the palpation method could replace Doppler in the identification of patients with severe vasculopathy [19]. In the general population, both methods were used separately to diagnose PAD in individuals with subclinical arteriosclerosis [16,20]. To the best of our knowledge, this is the first study in the clinical area that compares the use of Doppler and pulse palpation in identifying PAD and predicting CAD.

### 4.1. Usefulness of PAD to Predict CAD

We showed that palpation proved to be a simple, safe, and reliable method of detecting PAD when compared to Doppler. PAD diagnosed by both methods was useful to identify patients with a higher probability of CAD and, therefore, may help in the indication of invasive and non-invasive evaluation for CAD. In studies in high-risk patients with stage 5 CKD, it was observed that the presence of extracardiac atherosclerotic disease, such as PAD, was associated with a higher probability of CAD and a higher risk of cardiovascular events [3,15].

Confirming the assumption that patients with stage 5 CKD included in this study were indeed at high risk for CAD, the overall prevalence of coronary stenosis was 68.1%. The association of PAD with ischemic heart disease is frequent. This fact is due to the systemic nature of atherosclerosis, which usually affects multiple sites simultaneously [1]. Our results are in agreement with this concept, as patients with PAD and CAD had most of the risk factors in common. The exact prevalence of CAD in renal patients with PAD is poorly understood, but is likely to be at least as high as in the general population [4]. In the present series, we showed that CAD occurred in 80% of patients with PAD by the palpation method and in 79.1% by Doppler ultrasound. Both methods correlated significantly with CAD, with patients with PAD being twice as likely to have CAD. Studies with patients with stage 5 CKD reported that the prevalence of CAD in patients with PAD was 20% to 36% [7,20].

We observed that both the palpation and Doppler methods showed positive predictive values greater than 70%. For a test to be considered clinically useful, it must achieve sensitivity and specificity close to 80%. When the prevalence of the disease is high, as in our population, the sensitivity and, particularly, the negative predictive value must be even greater to prevent patients with severe disease from being undiagnosed [18]. In view of these findings, we understand that in our CKD population, PAD is associated with a high prevalence of CAD, but its absence did not exclude the need to indicate coronary angiography.

PAD diagnosed by the palpation method was also correlated with the prevalence of multivessel disease and with the possibility of indicating coronary intervention. This last result was, in our opinion, one of the most relevant in this study because it implies that the absence of pulses on palpation not only correlates with the anatomy, but allows us to suppose that subjects without peripheral pulses are more likely to be eligible for intervention even before angiography. Results similar to our study were found by Hur et al. [21], where 2687 patients who underwent preoperative coronary angiography before elective PAD were reviewed. The authors found that the prevalence of CAD in one or more epicardial arteries (>70% narrowing) was 55%. In another study, Cho et al. observed significant CAD (narrowing ≥ 50%) in 62% of patients with PAD undergoing peripheral and simultaneous coronary angiography [22].

### 4.2. Impact of PAD and CAD on Prognosis

We found that PAD confers a similar risk of future combined cardiovascular events and death from any cause in patients with stage 5 CKD. The influence of PAD remained significant even after adjustment for the interference of confounding factors. However, this increased risk was evidenced only when PAD was diagnosed by palpation of the pulses. The explanation of this apparent paradox is that the absence of a pulse probably functions as a marker of the severity and systemic spread of atherosclerotic disease. Therefore, it is understood that patients without pulses in the extremities, meaning severe PAD, should be considered at higher risk of events and death and treated as such, reinforcing the need for in-depth coronary investigation in this group. As it is well known that coronary events are common in this population, we expected that PAD would also be a predictor of myocardial infarction and unstable angina. This fact did not happen, however, probably because the number of coronary events was small in our sample.

Only PAD, diagnosed by the palpation method, was a predictor of combined cardiovascular events and death from any cause after the results were adjusted for other risk factors. In the literature, several risk factors associated with PAD in patients with CKD are mentioned [1,23]. In our study, the lack of impact of other classic risk factors on these outcomes may have been a consequence of the small number of patients in our sample. Alternatively, it could be considered that risk factors become relevant only when they cause major structural changes in the cardiovascular system.

### 4.3. Limitations

Our findings must be considered in the context of the following potential limitations. Given its observational design, selection bias and confounding may have impacted the observed relationships between identified predictors and our outcomes of interest. Another fact is that the study is a single center and had a small number of subjects included. Finally, because the patients included belong to a selected group, consequently, the results cannot be extrapolated to all individuals with CKD, especially for those who are not in stage 5 and on dialysis. Furthermore, we did not assess the impact of transplantation or coronary intervention on the event rate.

## 5. Conclusions

In conclusion, palpation of the pulses in lower limbs is an adequate, simple, safe, and inexpensive approach that can be used for the diagnosis of PAD, with no need for further evaluation in patients with stage 5 CKD. In patients with high cardiovascular risk, such as patients with CKD on dialysis, pulse palpation can guide the indication of invasive tests searching for coronary disease with a performance not inferior to myocardial scintigraphy. PAD, diagnosed by pulse palpation, is correlated with the occurrence of serious outcomes and identifies a group most likely to have an indication for coronary intervention. For these reasons, the detection of PAD by palpation can be considered an additional tool for risk stratification and can help in the selection of patients with an indication for invasive tests and, in this way, contribute to changing the current practices of clinical evaluation of patients on the waiting list for kidney transplantation.

## Figures and Tables

**Figure 1 jcm-12-05882-f001:**
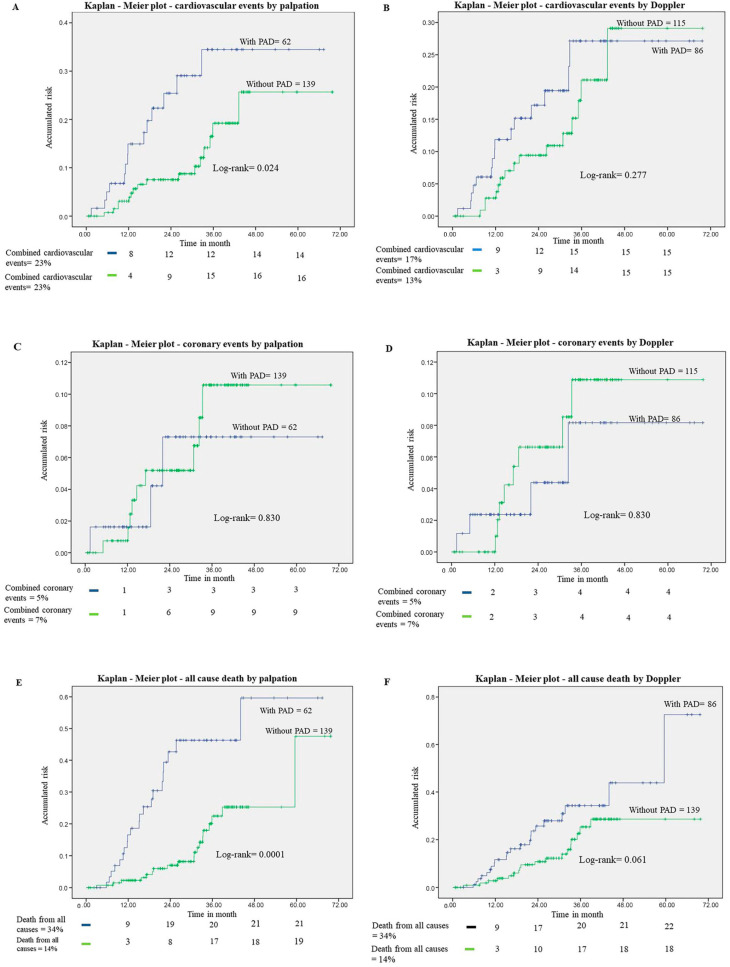
Kaplan–Meier plots. (**A**) Risk of combined cardiovascular events according to PAD by palpation method (*n* = 201). (**B**) Risk of combined cardiovascular events according to DAP by Doppler method (*n* = 201). (**C**) Risk of coronary events according to PAD by palpation method (*n* = 201). (**D**) Risk of coronary events according to DAP by Doppler method (*n* = 201). (**E**) Risk of death from any cause according to PAD by palpation method (*n* = 201). (**F**) Risk of death from any cause according to DAP by Doppler method (*n* = 201). PAD = peripheral arterial disease.

**Table 1 jcm-12-05882-t001:** Clinical characteristics of 201 patients with CKD on dialysis according to the presence of peripheral artery disease by absence of pulses by palpation.

Variable	Pulse by Palpation		*p*	Ultrasonography with Doppler		*p*	Total
	Absent *n* = 62	Present *n* = 139		Arteries with Obstruction ≥ 50% *n* = 86	Normal Arteries or Obstruction < 50% *n* = 115		
	*n* (%)	*n* (%)		*n* (%)	*n* (%)		*n* = 201
Age (years)	57.1 ± 10.9	54.4 ± 12.1	0.148 ^a^	58.9 ± 10.2	52.5 ± 12.1	<0.001	55.2 ± 11.8
Men	40 (64.5)	78 (56.1)	0.282	58 (67.4)	60 (52.2)	0.031	118 (58.7)
White	43 (69.4)	107 (77.0)	0.293	60 (69.8)	90 (78.3)	0.192	150 (74.6)
BMI Kg/m^2^ *	27.7 ± 4.8	26.9 ± 5.5	0.310	27.1 ± 4.7	27.2 ± 5.7	0.809 ^a^	27.1 ± 5.3
Dyslipidemia	22 (35.4)	43 (31)	0.515	31 (36.9)	34 (29.6)	0.288	65 (32.3)
Smoking	35 (56.5)	70 (50.4)	0.448	57 (49.6)	57 (49.6)	0.395	105 (52.2)
Diabetes	52 (83.9)	62 (44.6) *	<0.001	72 (83.7)	42 (36.5)	<0.001	114 (56.7)
Hypertension	34 (54.8)	82(59)	0.644	47 (54.7)	69 (60.0)	0.473	116 (57.7)
Stroke	9 (14.5)	10 (7.2)	0.120	9 (10.5)	10 (8.7)	0.808	19 (9.5)
AMI ^Ж^	10 (16.1)	19 (13.7)	0.667	9 (10.5)	20 (17.4)	0.224	29 (14.4)
Heart failure	4 (6.5)	18 (12.9)	0.224	6 (7)	16 (13.9)	0.170	22 (10.9)
LVMI ^ю^	114.8 ± 27.3	123.9 ± 43.6	0.094	113.4 ± 28.5	126.7 ± 45.1	0.079 ^a^	121.0 ± 39.4
LVPW ^ӧ^	10.8 ± 1.5	10.6 ± 1.6	0.512 ^a^	10.6 ± 1.5	10.7 ± 1.6	0.813 ^a^	10.7 ± 1.6
LVDD ^҂^	49.3 ± 6.5	49.5 ± 6.3	0.840 ^a^	49.4 ± 6.5	49.6 ± 6.2	0.697 ^a^	49.5 ± 6.3
LVEF ^¥^	0.6 ± 0.1	0.6 ± 0.1	0.233 ^a^	0.6 ± 0.1	0.6 ± 0.1	0.563 ^a^	0.6 ± 0.1
Hemodialysis ^∞^	28.6 ± 26.3	45.4 ± 51.3	0.038 ^a^	28.6 ± 26.7	49 ± 54.3	0.005 ^a^	40.2 ± 45.6
Creatinine	8.1 ± 3.3	8.4 ± 2.8	0.524	8.1 ± 3.3	8.6 ± 2.7	0.265	8.4 ± 3.0
T-cholesterol ^±^	158.1 ± 49.4	165.3 ± 46.8	0.200 ^a^	165.5 ± 47.6	161,3 ± 47.7	0.561 ^a^	163.1 ± 47.6
LDL ^ꝑ^	82.6 ± 39.9	86.9 ± 37.9	0.320 ^a^	85.7 ± 38.2	85.5 ± 38.9	0.992 ^a^	85.6 ± 38.5
HDL ^ꝩ^	45.8 ± 18.5	47.6 ± 14.8	0.148 ^a^	44.8 ± 17.7	48.6 ± 14.4	0.017 ^a^	47.0 ± 16.0
Triglycerides ^±^	156 ± 98.7	165.8 ± 110.5	0.693 ^a^	173.7 ± 128.1	154.9 ± 87.9	0.642 ^a^	162.8 ± 106.8

Note: ^a^ Non-parametric Mann–Whitney test—* Body mass index—^Ж^ Acute myocardial infarction—^ю^ Left ventricular mass index—^ӧ^ Left ventricular posterior wall—^҂^ Left ventricular diastolic diameters—^¥^ Left Ventricular ejection fraction—^∞^ Meses—^±^ mmol/L—^ꝑ^ Low-density lipoprotein—^ꝩ^ High-density lipoprotein.

**Table 2 jcm-12-05882-t002:** Specificity, sensitivity, positive predictive value, and negative predictive value of the methods for diagnosis of PAD in the detection of CAD.

Method	Specificity	Sensitivity	PPV ^a^	NPV ^b^
Palpation	76%	45%	80%	30%
USD Doppler	60%	66%	79%	43%

^a^ NPV = negative predictive value. ^b^ PPV = positive predictive value.

**Table 3 jcm-12-05882-t003:** Factors influencing the probability of combined cardiovascular events.

Variable	OR	95% CI	*p*-Value
Age ≥ 50 years	1.605	0.602–4.280	0.344
White race	1.751	0.667–4.492	0.259
Male sex	1.041	0.468–2.315	0.922
Smoking	0.494	0.225–1.084	0.079
Diabetes	0.653	0.280–1.525	0.325
Hypertension	1.464	0.674–3.183	0.335
PAD (palpation)	3.214	1.160–8.906	0.025
PAD (Doppler)	0.863	0.303–2.463	0.783

**Table 4 jcm-12-05882-t004:** Factors influencing the probability of death from any cause.

Variable	OR	95% CI	*p*-Value
Age ≥ 50 years	1.808	0.602–4.280	0.344
White race	1.741	0.667–4.492	0.259
Male sex	0.479	0.468–2.315	0.922
Smoking	0.988	0.506–1.977	0.997
Diabetes	1.475	0.697–3.124	0.310
Hypertension	0.469	0.240–0.916	0.027
PAD (Palpation)	2.653	1.158–6.080	0.021
PAD (Doppler)	0.816	0.352–1.891	0.635

## Data Availability

All the data used to support the findings in this study are included in the article.

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
