# Peer review of "Peripheral Artery Disease Diagnosed by Pulse Palpation as a Predictor of Coronary Artery Disease in Patients with Chronic Kidney Disease"

_jcm, 2023, doi:10.3390/jcm12185882_

Round 1
Reviewer 1 Report
Article “Peripheral Artery Disease Diagnosed by Pulse Palpation as a Predictor of Coronary Artery Disease in Patients with Chronic Kidney Disease.” deals with an important topic of diagnosing coronary artery disease in dialysis patients. This is a special group of patients at high risk of coronary artery disease. The authors attempted to diagnose CAD on the basis of PAD diagnosed by pulse palpation and Doppler examination. The clinical value of such a procedure is very limited, because both non-invasive and invasive diagnosis of CAD in dialysis patients is performed mainly in symptomatic patients and patients qualified for kidney transplantation.
My comments:
- Material and methods - the numbering of subsections should be corrected
- Should not patients with a history of myocardial infarction be excluded from the study?
- Statistical analysis says "Differences between groups were assessed with Fisher's exact test (for categorical data), or two-tailed Student's t test (for continuous data), or Mann-Whitney test for independent samples, when appropriate." I don't see such data in the results.
- Clinical data (presence of symptoms, BP and HR values, etc.), laboratory parameters, ECG and echocardiographic data are missing
- Table 2 most likely shows the results of the multivariate analysis. Please attach the results of the univariate analysis. It should be added by what method the analysis was performed and how the factors for the analysis were selected.
- In addition to Kaplan-Meier curves, the graphical representation of the results (tables and figures) can be improved.
Author Response
Thank you for your comments. 1. Comment: Material and methods - the numbering of subsections should be corrected Answer: Adequate 2. Comment: Should not patients with a history of myocardial infarction be excluded from the study? Answer: this is a good point. However, the proportion of patients with MI did not differ between groups. 3. Comment: Statistical analysis says "Differences between groups were assessed with Fisher's exact test (for categorical data), or two-tailed Student's t test (for continuous data), or Mann-Whitney test for independent samples, when appropriate." I don't see such data in the results. Answer: The cited analyzes are not in tables, but are mentioned in the text. 4. Comment: Clinical data (presence of symptoms, BP and HR values, etc.), laboratory parameters, ECG and echocardiographic data are missing Answer: We have these data, however, due to the restriction on the number of characters in the journal, we decided to focus on the association between PAD and CAD, the measures of performance of palpation and Doppler in relation to PAD and CAD, and the survival curves, as well as the associations with mortality. 5. Comment: Table 2 most likely shows the results of the multivariate analysis. Please attach the results of the univariate analysis. It should be added by what method the analysis was performed and how the factors for the analysis were selected. Answer: Logistic regression analysis was used to investigate the association of PAD/CAD and outcomes. To determine independent predictors of complications, the univariate binary logistic regression model was used, considering the clinical variables that could be associated with such outcomes. The multiple binary logistic regression model was applied using the ENTER method, including in the regression model the variables of clinical interest and with p–value <0.05 resulting from the univariate model, respecting the minimum limit of 10 events per variable included in the model. 6. In addition to Kaplan-Meier curves, the graphical representation of the results (tables and figures) can be improved. Answer: The numbering of the tables in the text were adapted, as well as the addition of Table 3. In this regard, we believe that the compilation of the Kaplan-Meier plots analyzes in a single figure can facilitate the visualization of the results. Despite this, we are open to suggestions by the journal's editorial.Reviewer 2 Report
I reviewed with interest the manuscript of Daniel Batista Conceição dos Santos et al. "Peripheral Artery Disease Diagnosed by Pulse Palpation as a Predictor of Coronary Artery Disease in Patients with Chronic Kidney Disease". In this article, the authors examine the possibility of using the diagnosis of Peripheral Artery Disease by Pulse Palpation to detect CAD and the risk of MACE in patients awaiting kidney transplantation on hemodialysis. The authors have shown that such a simple diagnostic test can be useful for both of these tasks. Probably, it can be useful for some categories of practitioners. However, during the review, I had the following comments and questions.
1. Why did the authors not use the ABI definition to detect Peripheral Artery Disease? This method is used in screening studies to detect Peripheral Artery Disease, is easy to use, does not require expensive equipment and is quite affordable in hemodialysis clinics.
2. The combination of severe Peripheral Artery Disease (and the absence of a pulse in one of the arteries of the foot indicates occlusion of one of the arteries) and CAD is well known. However, until now, it never occurred to anyone to diagnose CAD by the presence of Peripheral Artery Disease. I think that this article will not be an argument in favor of such a diagnostic strategy, given its low sensitivity and negative predictive value in the detection of CAD.
3. The article provides links to many old publications (more than 5 and 10 years old), but does not consider publications of recent years, in which the issues of diagnosing coronary pathology in patients with CKD are considered in sufficient detail (1,2). In addition, 2 reviews are devoted to this issue (3,4). By the way, the article by Kassab and Doukky (4) provides an algorithm for initial and continued CAD surveillance in kidney transplant candidates. I think that these algorithms are more applicable in clinical practice than those proposed by the authors of the peer-reviewed article.
4. Since there is no single point of view on the advisability or inadvisability of coronary revascularization in patients with end-stage CKD, coronary angiography and subsequent invasive treatment of coronary artery disease may be an important option in the treatment of a certain category of patients. This category of patients cannot be identified by Pulse Palpation.
5. The article does not contain Table 1 with the clinical characteristics of the included patients, although there is a link to it in the text. There are also no clinical data on patients (there are angina pectoris, symptoms of intermittent claudication, etc.). There are no data on the conduct / non-performing of revascularization and kidney transplantation. This is all the more surprising given that the study was conducted at a single center.
6. Table 3 is also missing, although it is mentioned in the text.
No comments
Author Response
Thank you for your thoughtful and relevant comments
- Comment: Why did the authors not use the ABI definition to detect Peripheral Artery Disease?
Answer: that is a good point. ABI is not usually used as a routine in clinical practice and palpation is. Therefore, a study based on palpation may have more practical relevance. Our patients were originally seen in a busy outpatient clinic and the investigators considered that a physical examination that included palpation of the main arteries was adequate to define the presence of vascular disease. Other tests were performed only when considered necessary. I believe that this practice is followed by the majority of services, although we agree that this work would be enriched by the inclusion of ABI..
- Comment: I think that this article will not be an argument in favor of such a diagnostic strategy, given its low sensitivity and negative predictive value in the detection of CAD
Answer: the objective of the work was to test a hypothesis. A negative result is also important, and point to the limitations of the method proposed but also showing its utility. For example, as we stated in Conclusions, pulse palpation can guide the indication of invasive tests searching for coronary disease with a performance not inferior to myocardial scintigraphy and correlated with the occurrence of serious outcomes. We feel that, because of its simplicity and no cost the utility of good physical exam should emphasized in the age dominated by frequently expensive technology.
- Comment: This category of patients cannot be identified by Pulse Palpation
Answer: we agree. The purpose of our work was not to determine the group of patients that would benefit from intervention. The issue of assess or not assess and to intervene or not intervene in patients with CAD and advanced CKD remains highly controversial and probably would only be clarified by larger prospective works. Herein we followed the international guidelines to decide on the modality of treatment. We believe that this is the best that could be done under the circumstance. On the other hand we did find an association between PAD diagnosed by palpation and the occurrence of multivessel coronary disease and with the probability of indication for coronary intervention. We are not recommending indicating coronary intervention based on the absence of peripheral pulse only that this finding may be helpful to identify patients more likely to benefit from intervention.
Adjustments were made to the missing tables.
The theoretical framework of the study was updated.
Reviewer 3 Report
Thank you for the opportunity to read the work entitled: Peripheral Artery Disease Diagnosed by Pulse Palpation as a Predictor of Coronary Artery Disease in Patients with Chronic Kidney Disease. The result presents an interesting perspective on the clinical assessment of patients with end-stage renal disease.
However, when reading the work, several questionable methodological assumptions draw attention, which undoubtedly translate into controversial conclusions. My main notes on the work are as follows:
- I would like to draw the authors' attention to the methodology of palpation of the pulse - it seems that such a very subjective examination, the results of which, in turn, are crucial from the perspective of the presented work, would probably require the involvement - as the authors rightly did in the assessment of coronary angiography - of at least two researchers and observed discrepancies between such studies would be an interesting element of the work.
- it seems to me that it is difficult to treat as equivalent definitions of the same clinical condition the absence of a palpable pulse and stenosis of at least 50% in the USD study. Thus I am afraid that in the further part of the work, we are trying to compare apples to pears
- taking into account the above comments, it seems that the conclusion: The risk of combined serious CV events and death was significantly higher in subjects with PAD diagnosed by palpation but not by USD is not justified in the presented data due to the adopted definition of PAD
- conclusion: palpation of the pulses in lower limbs is an adequate, simple, safe, and inexpensive approach that can be used for the diagnosis of PAD, with no need for further evacuation in patients with stage 5 CKD does not seem to be justified in the presented results
- conclusion In patients with high cardiovascular risk, such as patients
with CKD under dialysis, pulse palpation can guide the indication of invasive tests searching for coronary disease with a performance not inferior to myocardial scintigraphy is certainly not justified by the presented results. Moreover, I am afraid that it is in opposition to current medical knowledge
What I would like to propose finally, taking into account the effort of data collection and the information gathered, is to consider finding a cut-off point for the advancement of atherosclerosis in the ultrasound examination that best correlates with the results of long-term follow-up and then checking how the results of the pulse palpation examination are distributed in groups defined in this way.
Author Response
We greatly appreciated your comments.
- Comment: …would probably require the involvement - as the authors rightly did in the assessment of coronary angiography - of at least two researchers.
Answer: it is correct. The exam was performed always by the same investigator at least 2 times: at the beginning and at the end of the exam. We admit that this is a limitation.
- Comment: difficult to treat as equivalent definitions of the same clinical condition the absence of a palpable pulse and stenosis of at least 50% in the USD study.
Answer: right. But the alternative would be performing a more expensive and invasive procedure in all of our patients. We accept that the absence of pulse and stenosis of at least 50% in the USD study are not exactly equivalent but both mean that blood flow is severely compromised and that this is clinically relevant. Notwithstanding, we agree to withdraw the statement you mentioned from the manuscript.
- Comment: The risk of combined serious CV events and death was significantly higher in subjects with PAD diagnosed by palpation but not by USD.
Answer: this was a finding; it is derived from data. The interpretation is certainly open to dispute.
- Commnent: with no need for further evaluation in patients with stage 5 CKD.
The statement was withdrawn.
- Comment: I am afraid that it is in opposition to current medical knowledge.
Answer: that is possibly true but I believe that does not disqualify the statement. As far as I know there is no investigation comparing the 2 methods in this setting.
- Comment: What I would like to propose.
Answer: that should be the best thing to do. Indeed, we considered that but the USD we have was incomplete in that regard not allowing any further analysis.
Round 2
Reviewer 1 Report
Dear Authors
Thanks for the answers. For the future, I suggest marking changes made in the manuscript, e.g. with a different font color.
I have no more comments.
Kind regards
Author Response
Thanks a lot for the notes. I made the markings in the text as instructed.
Reviewer 2 Report
The authors of the manuscript responded in part to my comments, including 2 references to recent studies. However, the answer to the third remark did not satisfy me. I encourage authors to consider the following links:
11. Kanigicherla DAK, Bhogal T, Stocking K, Chinnadurai R, Gray S, Javed S, Fortune C, Augustine T, Kalra PA. Non-invasive cardiac stress studies may not offer significant benefit in pre-kidney transplant evaluation: A retrospective cohort study. PLoS One. 2020 Oct 28;15(10):e0240912. doi: 10.1371/journal.pone.0240912.
22. Steinmetz T, Perl L, Zvi BR, Atamna M, Kornowski R, Shiyovich A, Hamdan A, Nesher E, Rahamimov R, Gal TB, Skalsky K. The prognostic value of pre-operative coronary evaluation in kidney transplanted patients. Front Cardiovasc Med. 2022 Aug 5;9:974158. doi: 10.3389/fcvm.2022.974158.
33. Lioudaki E, Androvitsanea A, Petrakis I, Bakogiannis C, Androulakis E. Cardiac Imaging and Management of Cardiac Disease in Asymptomatic Renal Transplant Candidates: A Current Update. Diagnostics (Basel). 2022 Sep 27;12(10):2332. doi: 10.3390/diagnostics12102332.
44. Kassab K, Doukky R. Cardiac imaging for the assessment of patients being evaluated for kidney transplantation. J Nucl Cardiol. 2022 Apr;29(2):543-557. doi: 10.1007/s12350-021-02561-6.
No comments
Author Response
From the requested suggestions, we made adjustments in the introduction of the manuscript.